# Diverse Shape Completion via Style Modulated Generative Adversarial Networks

**Wesley Khademi**
Oregon State University
khademiw@oregonstate.edu

**Li Fuxin**
Oregon State University
lif@oregonstate.edu

## Abstract

Shape completion aims to recover the full 3D geometry of an object from a partial observation. This problem is inherently multi-modal since there can be many ways to plausibly complete the missing regions of a shape. Such diversity would be indicative of the underlying uncertainty of the shape and could be preferable for downstream tasks such as planning. In this paper, we propose a novel conditional generative adversarial network that can produce many diverse plausible completions of a partially observed point cloud. To enable our network to produce multiple completions for the same partial input, we introduce stochasticity into our network via style modulation. By extracting style codes from complete shapes during training, and learning a distribution over them, our style codes can explicitly carry shape category information leading to better completions. We further introduce diversity penalties and discriminators at multiple scales to prevent conditional mode collapse and to train without the need for multiple ground truth completions for each partial input. Evaluations across several synthetic and real datasets demonstrate that our method achieves significant improvements in respecting the partial observations while obtaining greater diversity in completions.

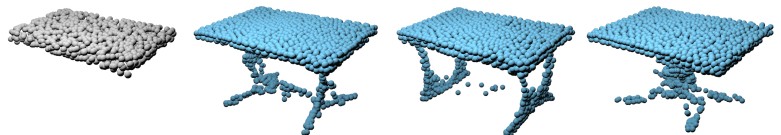

Figure 1: Given a partially observed point cloud (gray), our method is capable of producing many plausible completions (blue) of the missing regions.

## 1 Introduction

With the rapid advancements in 3D sensing technologies, point clouds have emerged as a popular representation for capturing the geometry of real-world objects and scenes. Point clouds come from sensors such as LiDAR and depth cameras, and can find applications in various domains such as robotics, computer-aided design, augmented reality, and autonomous driving. However, the 3D geometry produced by such sensors is typically sparse, noisy, and incomplete, which hinders their effective utilization in many downstream tasks.

This motivates the task of 3D shape completion from a partially observed point cloud, which has seen significant research in the past few years [1, 2, 3, 4, 5, 6, 7]. Many early point cloud completion works mostly focus on generating a single completion that matches the ground truth in the training set, which does not take into account the potential uncertainty underlying the complete point cloud given the partial view. Ideally, an approach should correctly characterize such uncertainty – generating mostly similar completions when most of the object has been observed, and less similar completions when less of the object has been observed. A good characterization of uncertainty would be informative for downstream tasks such as planning or active perception to aim to reduce such uncertainty.

37th Conference on Neural Information Processing Systems (NeurIPS 2023).

The task of completing a 3D point cloud with the shape uncertainty in mind is called *multi-modal shape completion* [8, 9], which aims to generate diverse point cloud completions (not to be confused with multi-modality of the input, e.g. text+image). A basic idea is to utilize the diversity coming from generative models such as generative adversarial networks (GAN), where diversity, or the avoidance of *mode collapse* to always generate the same output, has been studied extensively. However, early works [8, 9] often obtain diversity at a cost of poor fidelity to the partial observations due to their simplistic completion formulation that decodes from a single global latent vector. Alternatively, recent diffusion-based methods [10, 11, 12] and auto-regressive methods [13, 14] have shown greater generation capability, but suffer from slow inference time.

In this paper, we propose an approach to balance the diversity of the generated completions and fidelity to the input partial points. Our first novelty comes from the introduction of a *style encoder* to encode the global shape information of complete objects. During training, the ground truth shapes are encoded with this style encoder so that the completions match the ground truth. However, multiple ground truth completions are not available for each partial input; therefore, only using the ground truth style is not enough to obtain diversity in completions. To overcome this, we randomly sample style codes to provide diversity in the other generated completions. Importantly, we discovered that **reducing** the capacity of the style encoder and adding noise to the encoded ground truth shape leads to improved diversity of the generated shapes. We believe this avoids the ground truth from encoding too much content, which may lead the model to overfit to only reconstructing ground truth shapes.

Besides the style encoder, we also take inspiration from recent work SeedFormer [7] to adopt a coarse-to-fine completion architecture. SeedFormer has shown high-quality shape completion capabilities with fast inference time, but only provides deterministic completions. In our work, we make changes to the layers of SeedFormer, making it more suitable for the multi-modal completion task. Additionally, we utilize discriminators at **multiple scales**, which enable training without multiple ground truth completions and significantly improves completion quality. We further introduce a multi-scale diversity penalty that operates in the feature space of our discriminators. This added regularization helps ensure different sampled style codes produce diverse completions.

With these improvements, we build a multi-modal point cloud completion algorithm that outperforms state-of-the-art in both the fidelity to the input partial point clouds as well as the diversity in the generated shapes. Our method is capable of fast inference speeds since it does not rely on any iterative procedure, making it suitable for real-time applications such as in robotics.

Our main contributions can be summarized as follows:

- We design a novel conditional GAN for the task of diverse shape completion that achieves greater diversity along with higher fidelity partial reconstruction and completion quality.

- We introduce a style-based seed generator that produces diverse coarse shape completions via style modulation, where style codes are learned from a distribution of complete shapes.

- We propose a multi-scale discriminator and diversity penalty for training our diverse shape completion framework without access to multiple ground truth completions per partial input.

## 2  Related work

### 2.1  3D shape generation

The goal of 3D shape generation is to learn a generative model that can capture the distribution of 3D shapes. In recent years, generative modeling frameworks have been investigated for shape generation using different 3D representations, including voxels, point clouds, meshes, and neural fields.

One of the most popular frameworks for 3D shape generation has been generative adversarial networks (GANs). Most works have explored point cloud-based GANs [15, 16, 17, 18, 19], while recent works have shown that GANs can be trained on neural representations [20, 21, 22, 23]. To avoid the training instability and mode collapse in GANs, variational autoencoders have been used for learning 3D shapes [24, 25, 26], while other works have made use of Normalizing Flows [27, 28, 29]. Recently, diverse shape generation has been demonstrated by learning denoising diffusion models on point clouds [30], latent representations [31, 32], or neural fields [33].

## 2.2 Point cloud completion

Point cloud completion aims to recover the missing geometry of a shape while preserving the partially observed point cloud. PCN [1] was among the earliest deep learning-based methods that worked directly on point clouds using PointNet [34] for point cloud processing. Since then, other direct point cloud completion methods [35, 36, 4, 37, 6, 38, 7] have improved completion results by using local-to-global feature extractors and by producing completions through hierarchical decoding.

Recently, SeedFormer [7] has achieved state-of-the-art performance in point cloud completion. Their Patch Seeds representation has shown to be more effective than global shape representations due to carrying learned geometric features and explicit shape structure. Furthermore, their transformer-based upsampling layers enable reasoning about spatial relationships and aggregating local information in a coarse-to-fine manner, leading to improved recovery of fine geometric structures.

GAN-based completion networks have also been studied to enable point cloud completion learning in unpaired [39, 40, 41] or unsupervised [42] settings. To enhance completion quality, some works have leveraged adversarial training alongside explicit reconstruction losses [4, 43].

## 2.3 Multimodal shape completion

Most point cloud completion models are deterministic despite the ill-posed nature of shape completion. To address this, Wu et al. [8] proposed a GAN framework that learns a stochastic generator, conditioned on a partial shape code and noise sample, to generate complete shape codes in the latent space of a pre-trained autoencoder. Arora et al. [9] attempt to mitigate the mode collapse present in [8] by using implicit maximum likelihood estimation. These methods can only represent coarse geometry and struggle to respect the partial input due to decoding from a global shape latent vector.

ShapeFormer [13] and AutoSDF [14] explore auto-regressive approaches for probabilistic shape completion. Both methods propose compact discrete 3D representations for shapes and learn an auto-regressive transformer to model the distribution of object completions on such representation. However, these methods have a costly sequential inference process and rely on voxelization and quantization steps, potentially resulting in a loss of geometric detail.

Zhou et al. [10] propose a conditional denoising diffusion model that directly operates on point clouds to produce diverse shape completions. Alternatively, DiffusionSDF [11] and SDFusion [12] first learn a compact latent representation of neural SDFs and then learn a diffusion model over this latent space. These methods suffer from slow inference times due to the iterative denoising procedure, while [11, 12] have an additional costly dense querying of the neural SDF for extracting a mesh.

## 2.4 Diversity in GANs

Addressing diversity in GANs has also been extensively studied in the image domain. In particular, style-based generators have shown impressive capability in high-quality diverse image generation where several methods have been proposed for injecting style into generated images via adaptive instance normalization [44, 45, 46, 47] or weight modulation [46, 48]. In the conditional setting, diversity has been achieved by enforcing invertibility between output and latent codes [49] or by regularizing the generator to prevent mode collapse [50, 51, 52] .

# 3 Method

In this section, we present our conditional GAN framework for diverse point cloud completion. The overall architecture of our method is shown in Figure 2.

Our generator is tasked with producing high-quality shape completions when conditioned on a partial point cloud. To accomplish this, we first introduce a new partial shape encoder which extracts features from the partial input. We then follow SeedFormer [7] and utilize a seed generator to first propose a sparse set of points that represent a coarse completion of a shape given the extracted partial features. The coarse completion is then passed through a series of upsampling layers that utilize transformers with local attention to further refine and upsample the coarse completion into a dense completion.

To obtain diversity in our completions, we propose a style-based seed generator that introduces stochasticity into our completion network at the coarsest level. Our style-based seed generator

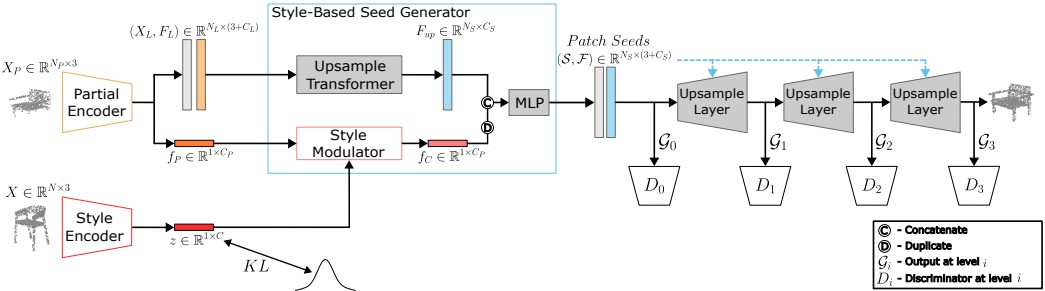

Figure 2: Overview of our diverse shape completion framework. A partial encoder is used to extract information from a partial point cloud. During training, a style encoder extracts style codes from complete point clouds, and at inference time style codes are randomly sampled from a normal distribution. Sampled style codes are injected into the partial information to produce diverse Patch Seeds in our style-based seed generator. The generated Patch Seeds are then upsampled into a dense completion through upsampling layers. Furthermore, discriminators and diversity penalties are used at every upsampling layer to train our model.

modulates the partial shape information with style codes before producing a sparse set of candidate points, enabling diverse coarse shape completions that propagate to dense completions through upsampling layers. The style codes used in modulating partial shape information are learned from an object category's complete shapes via a style encoder. Finally, we introduce discriminators and diversity penalties at multiple scales to train our model to produce diverse high-quality completions without having access to multiple ground truth completions that correspond to a partial observation.

## 3.1 Partial shape encoder

The goal of our partial encoder is to extract shape information in a local-to-global fashion, extracting local information that will be needed in the decoding stage to reconstruct fine geometric structures, while capturing global information needed to make sure a globally coherent shape is generated. An overview of the architecture for our proposed partial encoder is shown in Figure 3a.

Our partial encoder takes in a partially observed point cloud $X_P$ and first applies a MLP to obtain a set of point-wise features $F_0$. To extract shape information in a local-to-global fashion, $L$ consecutive downsampling blocks are applied to obtain a set of downsampled points $X_L$ with local features $F_L$. In each downsampling block, a grid downsampling operation is performed followed by a series of PointConv [53] layers for feature interpolation and aggregation. Following the downsampling blocks, a global representation of the partial shape is additionally extracted by an MLP followed by max pooling, producing partial latent vector $f_P$.

## 3.2 Style encoder

To produce multi-modal completions, we need to introduce randomness into our completion model. One approach is to draw noise from a Gaussian distribution and combine it with partial features during the decoding phase. Another option is to follow StyleGAN [46, 54] and transform noise samples through a non-linear mapping to a latent space $\mathcal{W}$ before injecting them into the partial features. However, these methods rely on implicitly learning a connection between latent samples and shape information. Instead, we propose to learn style codes from an object category's distribution of complete shapes and sample these codes to introduce stochasticity in our completion model. Our style codes explicitly carry information about ground truth shapes, leading to higher quality and more diverse completions.

To do so, we leverage the set of complete shapes we have access to during training. We introduce an encoder $E$ that maps a complete shape $X \in \mathbb{R}^{N \times 3}$ to a global latent vector via a 4-layer MLP followed by max pooling. We opt for a simple architecture as we would like the encoder to capture high level information about the distribution of shapes rather than fine-grained geometric structure.

Instead of assuming we have complete shapes to extract style codes from at inference time, we learn a distribution over style codes that we can sample from. Specifically, we define our style encoder as a learned Gaussian distribution $E_S(z|X) = \mathcal{N}(z|\mu(E(X)), \sigma(E(X)))$ by adding two fully connected layers, $\mu$ and $\sigma$, to the encoder $E$ to predict the mean and standard deviation of a Gaussian to sample a style code from. Since our aim is to learn style codes that convey information about complete shapes that is useful for generating diverse completions, we train our style encoder with guidance from our

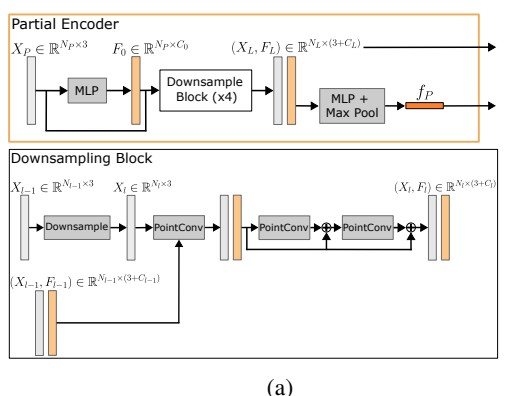
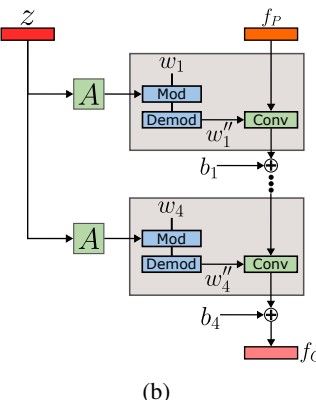

|  (a)  |  (b)  |

Figure 3: (a) Architecture of our partial shape encoder. (b) Overview of our style modulator network. For each style-modulated convolution (gray box), $w_i$ and $b_i$ are learned weights and biases of a convolution, $w_i''$ are the weights after the modulation and demodulation process, and $A$ is a learned Affine transformation.

completion network's losses. To enable sampling during inference, we minimize the KL-divergence between $E_S(z|X)$ and a normal distribution during training. We additionally find that adding noise to our sampled style codes during training leads to higher fidelity and more diverse completions.

### 3.3 Style-based seed generator

We make use of the Patch Seeds representation proposed in SeedFormer [7], which enables faithfully completing unobserved regions while preserving partially observed structures. Patch Seeds are defined as a set of seed coordinates $\mathcal{S} \in \mathbb{R}^{N_S \times 3}$ and seed features $\mathcal{F} \in \mathbb{R}^{N_S \times C_S}$ produced by a seed generator. In particular, a set of upsampled features $F_{up} \in \mathbb{R}^{N_S \times C_S}$ are generated from the partial local features $(X_L, F_L)$ via an Upsample Transformer [7]. Seed coordinates $\mathcal{S}$ and features $\mathcal{F}$ are then produced from upsampled features $F_{up}$ concatenated with partial latent code $f_P$ via an MLP.

However, the described seed generator is deterministic, prohibiting the ability to generate diverse Patch Seeds that can then produce diverse completions through upsampling layers. We propose to incorporate stochasticity into the seed generator by injecting stochasticity into the partial latent vector $f_P$. We introduce a style modulator network $M(f_P, z)$, shown in Figure 3b, that injects a style code $z$ into a partial latent vector $f_P$ to produce a styled partial shape latent vector $f_C$. Following [54], we use weight modulation to inject style into the activation outputs of a network layer, where the demodulated weights $w''$ used in each convolution layer are computed as:

$$s = A(z), \text{mod}: w'_{ijk} = s_i \cdot w_{ijk}, \text{demod}: w''_{ijk} = w'_{ijk} \Big/ \sqrt{\sum_{i,k} w'^2_{ijk} + \epsilon} \tag{1}$$

where $A$ is an Affine transformation, and $w$ is the original convolution weights with $i, j, k$ corresponding to the input channel, output channel, and spatial footprint of the convolution, respectively.

### 3.4 Coarse-to-fine decoder

Our decoder operates in a coarse-to-fine fashion, which has been shown to be effective in producing shapes with fine geometric structure for the task of point cloud completion [36, 38, 7, 4]. We treat our Patch Seed coordinates $\mathcal{S}$ as our coarsest completion $\mathcal{G}_0$ and progressively upsample by a factor $r$ to produce denser completions $\mathcal{G}_i$ $(i = 1, ..., 3)$. At each upsampling stage $i$, a set of seed features are first interpolated from the Patch Seeds. Seed features along with the previous layer's points and features are then used by an Upsample Transformer [7] where local self-attention is performed to produce a new set of points and features upsampled by a factor $r$. We replace the inverse distance weighted averaging used to interpolate seed features from Patch Seeds in SeedFormer with a PointConv interpolation. Since the importance of Patch Seed information may vary across the different layers, we believe a PointConv interpolation is more appropriate than a fixed weighted interpolation as it can learn the appropriate weighting of Patch Seed neighborhoods for each upsampling layer.

Unlike the fully-connected decoders in [8, 9], the coarse-to-fine decoder used in our method can reason about the structures of local regions, allowing us to generate cleaner shape surfaces. A coarse-to-fine design provides us with the additional benefit of having discriminators and diversity penalties at multiple resolutions, which we have found to lead to better completion fidelity and diversity.

## 3.5 Multi-scale discriminator

During training, we employ adversarial training to assist in learning realistic completions for any partial input and style code combination. We introduce a set of discriminators $D_i$ for $i = \{0, ..., 3\}$, to discriminate against real and fake point clouds at each output level of our generator. Each discriminator follows a PointNet-Mix architecture proposed by Wang et al. [55]. In particular, an MLP first extracts a set of point features from a shape, which are max-pooled and average-pooled to produce $f_{max}$ and $f_{avg}$, respectively. The features are then concatenated to produce *mix-pooled feature* $f_{mix} = [f_{max}, f_{avg}]$ before being passed through a fully-connected network to produce a final score about whether the point cloud is real or fake.

We also explored more complex discriminators that made use of PointConv or attention mechanisms, but we were unable to successfully train with any such discriminator. This is in line with the findings in [55], suggesting that more powerful discriminators may not guide the learning of point cloud shape generation properly. Thus, we instead use a weaker discriminator architecture but have multiple of them that operate at different scales to discriminate shape information at various feature levels.

For training, we use the WGAN loss [56] with R1 gradient penalty [57] and average over the losses at each output level $i$. We let $\hat{X}_i = \mathcal{G}_i(X_P, z)$ be the completion output at level $i$ for partial input $X_P$ and sampled style code $z \sim E_S(z|X)$, and let $X_i$ be a real point cloud of same resolution. Then our discriminator loss $\mathcal{L}_D$ and generator loss $\mathcal{L}_G$ are defined as:

$$\mathcal{L}_D = \frac{1}{4} \sum_{i=0}^{3} \left( \mathbb{E}_{\hat{X} \sim P(\hat{X})} \left[ D_i(\hat{X}_i) \right] - \mathbb{E}_{X \sim P(X)} [D_i(X_i)] + \frac{\gamma}{2} \mathbb{E}_{X \sim P(X)} \left[ \|\nabla D_i(X_i)\|^2 \right] \right) \quad (2)$$

$$\mathcal{L}_G = -\frac{1}{4} \sum_{i=0}^{3} \left( \mathbb{E}_{\hat{X} \sim P(\hat{X})} [D_i(\hat{X}_i)] \right) \quad (3)$$

where $\gamma$ is a hyperparameter ($\gamma = 1$ in our experiments), $P(\hat{X})$ is the distribution of generated shapes, and $P(X)$ is the distribution of real shapes.

## 3.6 Diversity regularization

Despite introducing stochasticity into the partial latent vector, it is still possible for the network to learn to ignore the style code $z$, leading to mode collapse to a single completion. To address this, we propose a diversity penalty that operates in the feature space of our discriminator. Our key insight is that for a discriminator to be able to properly discriminate between real and fake point clouds, its extracted features should have learned relevant structural information. Then our assumption is that if two completions are structurally different, the discriminator's global mix-pooled features should be dissimilar as well, which we try to enforce through our diversity penalty.

Specifically, at every training iteration we sample two style codes $z_1 \sim E_S(z|X_1)$ and $z_2 \sim E_S(z|X_2)$ from random complete shapes $X_1$ and $X_2$. For a single partial input $X_P$, we produce two different completions $\mathcal{G}_i(X_P, z_1)$ and $\mathcal{G}_i(X_P, z_2)$. We treat our discriminator $D_i$ as a feature extractor and extract the mixed-pooled feature for both completions at every output level $i$. We denote the mixed-pooled feature corresponding to a completion conditioned on style code $z$ at output level $i$ by $f_{mix_i}^z$, then minimize:

$$\mathcal{L}_{div} = \sum_{i=0}^{3} \frac{1}{\left\| f_{mix_i}^{z_1} - f_{mix_i}^{z_2} \right\|_1} \quad (4)$$

which encourages the generator to produce completions with dissimilar mix-pooled features for different style codes. Rather than directly using the discriminator's mix-pooled feature, we perform pooling only over the set of point features that are not in partially observed regions. This helps avoid penalizing a lack of diversity in the partially observed regions of our completions.

We additionally make use of a partial reconstruction loss at each output level on both completions:

$$\mathcal{L}_{part} = \sum_{z \in \{z_1, z_2\}} \sum_{i=0}^{3} d^{UHD}(X_P, \mathcal{G}_i(X_P, z)) \quad (5)$$

where $d^{UHD}$ stands for the unidirectional Hausdorff distance from partial point cloud to completion. Such a loss helps ensure that our completions respect the partial input for any style code $z$.

To ensure that the completion set covers the ground truth completions in the training set, we choose to always set random complete shape $X_1 = X_{GT}$ and sample $z_1 \sim E_S(z|X_{GT})$, where $X_{GT}$ is the corresponding ground truth completion to the partial input $X_P$. This allows us to provide supervision at the output of each upsampling layer via Chamfer Distance (CD) for one of our style codes:

$$\mathcal{L}_{comp} = \sum_{i=0}^{3} d^{CD}(X_{GT}, \ \mathcal{G}_i(X_P, z_1)) \tag{6}$$

Our full loss that we use in training our generator is then:

$$\mathcal{L} = \lambda_G \mathcal{L}_G + \lambda_{comp} \mathcal{L}_{comp} + \lambda_{part} \mathcal{L}_{part} + \lambda_{div} \mathcal{L}_{div} \tag{7}$$

We set $\lambda_G = 1, \lambda_{comp} = 0.5, \lambda_{part} = 1, \lambda_{div} = 5$, which we found to be good default settings across the datasets used in our experiments.

## 4 Experiments

In this section, we evaluate our method against a variety of baselines on the task of multimodal shape completion and show superior quantitative and qualitative results across several synthetic and real datasets. We further conduct a series of ablations to justify the design choices of our method.

**Implementation Details** Our model takes in $N_P = 1024$ points as partial input and produces $N = 2048$ points as a completion. For training the generator, the Adam optimizer is used with an initial learning rate of $1 \times 10^{-4}$ and the learning rate is linearly decayed every 2 epochs with a decay rate of 0.98. For the discriminator, the Adam optimizer is used with a learning rate of $1 \times 10^{-4}$. We train a separate model for each shape category and train each model for 300 epochs with a batch size of 56. All models are trained on two NVIDIA Tesla V100 GPUs and take about 30 hours to train.

**Datasets** We conduct experiments on several synthetic and real datasets. Following the setup of [8], we evaluate our approach on the Chair, Table, and Airplane categories of the 3D-EPN dataset [58]. Similarly, we also perform experiments on the Chair, Table, and Lamp categories from the PartNet dataset [59]. To evaluate our method on real scanned data, we conduct experiments on the Google Scanned Objects (GSO) dataset [60]. For GSO, we share quantitative and qualitative results on the Shoe, Toys, and Consumer Goods categories. A full description is presented in the supplementary.

**Metrics** We follow [8] and evaluate with the Minimal Matching Distance (MMD), Total Mutual Difference (TMD), and Unidirectional Hausdorff Distance (UHD) metrics. MMD measures the fidelity of the completion set with respect to the ground truth completions. TMD measures the completion diversity for a partial input shape. UHD measures the completion fidelity with respect to the partial input. We evaluate metrics on $K = 10$ generated completions per partial input. Reported MMD, TMD, and UHD values in our results are multiplied by $10^3$, $10^2$, and $10^2$, respectively.

Table 1: Results on the 3D-EPN dataset. * indicates metric is not reported.

| Method | MMD ↓ | | | | TMD ↑ | | | | UHD ↓ | | | |
|---|---|---|---|---|---|---|---|---|---|---|---|---|
| | Chair | Plane | Table | Avg. | Chair | Plane | Table | Avg. | Chair | Plane | Table | Avg. |
| SeedFormer [7] | 0.45 | 0.17 | 0.65 | 0.42 | 0.00 | 0.00 | 0.00 | 0.00 | 1.69 | 1.27 | 1.69 | 1.55 |
| KNN-latent [8] | 1.45 | 0.93 | 2.25 | 1.54 | 2.24 | 1.13 | 3.25 | 2.21 | 8.94 | 9.54 | 12.70 | 10.39 |
| cGAN [8] | 1.61 | 0.82 | 2.57 | 1.67 | 2.56 | 2.03 | 4.49 | 3.03 | 8.33 | 9.59 | 9.03 | 8.98 |
| IMLE [9] | * | * | * | * | 2.93 | **2.31** | 4.92 | **3.39** | 8.51 | 9.55 | 8.52 | 8.86 |
| Ours | **1.16** | **0.59** | **1.45** | **1.07** | **3.26** | 1.53 | **5.14** | 3.31 | **4.02** | **3.40** | **4.00** | **3.81** |

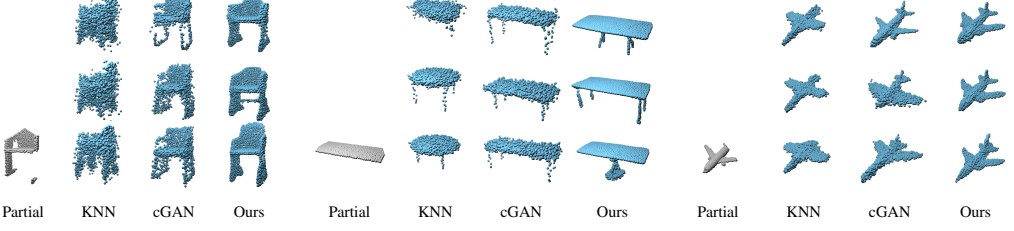

Figure 4: Qualitative comparison of multi-modal completions on the 3D-EPN dataset.

Table 2: Results on the PartNet dataset. * indicates that metric is not reported. † indicates methods that use an alternative computation for MMD and TMD.

| Method | MMD ↓ | | | | TMD ↑ | | | | UHD ↓ | | | |
|---|---|---|---|---|---|---|---|---|---|---|---|---|
| | Chair | Lamp | Table | Avg. | Chair | Lamp | Table | Avg. | Chair | Lamp | Table | Avg. |
| SeedFormer [7] | 0.72 | 1.35 | 0.71 | 0.93 | 0.00 | 0.00 | 0.00 | 0.00 | 1.54 | 1.25 | 1.48 | 1.42 |
| KNN-latent [8] | **1.39** | **1.72** | 1.30 | 1.47 | 2.28 | 4.18 | 2.36 | 2.94 | 8.58 | 8.47 | 7.61 | 8.22 |
| cGAN [8] | 1.52 | 1.97 | 1.46 | 1.65 | 2.75 | 3.31 | 3.30 | 3.12 | 6.89 | 5.72 | 5.56 | 6.06 |
| IMLE [9] | * | * | * | * | 2.76 | 5.49 | 4.45 | 4.23 | 6.17 | 5.58 | 5.16 | 5.64 |
| ShapeFormer [10] | * | * | * | **1.32** | * | * | * | 3.96 | * | * | * | * |
| Ours | 1.50 | 1.84 | **1.15** | 1.49 | **4.36** | **6.55** | **5.11** | **5.34** | **3.79** | **3.88** | **3.69** | **3.79** |
| PVD [10] † | **1.27** | **1.03** | 1.98 | 1.43 | 1.91 | 1.70 | **5.92** | 3.18 | * | * | * | * |
| Ours † | 1.34 | 1.55 | **1.12** | **1.34** | **5.27** | **7.11** | 5.84 | **6.07** | * | * | * | * |

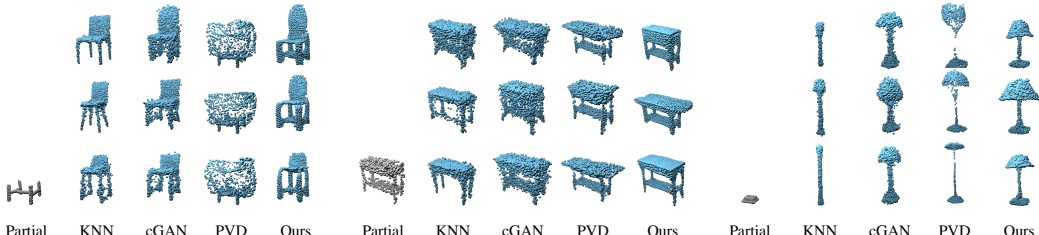

Figure 5: Qualitative results on the PartNet dataset.

**Baselines** We compare our model against three direct multi-modal shape completion methods: cGAN [8], IMLE [9], and KNN-latent which is a baseline proposed in [8]. We further compare with the diffusion-based method PVD [10] and the auto-regressive method ShapeFormer [13]. We also share quantitative results against the deterministic point cloud completion method SeedFormer [7].

## 4.1 Results

Results on the 3D-EPN dataset are shown in Table 1. SeedFormer obtains a low UHD implying that their completions respect the partial input well; however, their method produces no diversity as it is deterministic. Our UHD is significantly better than all multi-modal completion baselines, suggesting that we more faithfully respect the partial input. Additionally, our method outperforms others in terms of TMD and MMD, indicating better diversity and completion quality. This is also reflected in the qualitative results shown in Figure 4, where KNN-latent fails to produce plausible completions, while completions from cGAN contain high levels of noise.

In Table 2, we compare against other methods on the PartNet dataset. For a fair comparison with PVD, we also report metrics following their protocol (denoted by †)[10]. In particular, under their protocol TMD is computed on a subsampled set of 1024 points and MMD is computed on the subsampled set concatenated with the original partial input. Once again our method obtains the best diversity (TMD) across all categories, beating out the diffusion-based method PVD and the auto-regressive method ShapeFormer. Our method also achieves significantly lower UHD and shows competitive performance in terms of MMD. Some qualitative results are shown in Figure 5. We find that our method produces cleaner surface geometry and obtains nice diversity in comparison to other methods.

Table 3: Results on Google Scanned Objects dataset. * indicates metric is not reported. † indicates methods that use an alternative computation for MMD and TMD.

| Method | MMD ↓ | | | | TMD ↑ | | | | UHD ↓ | | | |
|---|---|---|---|---|---|---|---|---|---|---|---|---|
| | Shoe | Toys | Goods | Avg. | Shoe | Toys | Goods | Avg. | Shoe | Toys | Goods | Avg. |
| SeedFormer [7] | 0.42 | 0.67 | 0.47 | 0.52 | 0.00 | 0.00 | 0.00 | 0.00 | 1.49 | 1.69 | 1.55 | 1.58 |
| cGAN [8] | 1.00 | 2.75 | 1.79 | 1.85 | 1.10 | 1.87 | **1.95** | 1.64 | 5.05 | 6.61 | 6.35 | 6.00 |
| Ours | **0.85** | **1.90** | **0.99** | **1.25** | **1.71** | **2.27** | 1.89 | **1.96** | **2.88** | **3.84** | **4.68** | **3.80** |
| PVD [10] † | **0.66** | 2.04 | 1.11 | 1.27 | 1.15 | 2.05 | 1.44 | 1.55 | * | * | * | * |
| Ours † | 0.90 | **1.72** | **1.04** | **1.22** | **2.42** | **2.88** | **2.57** | **2.62** | * | * | * | * |

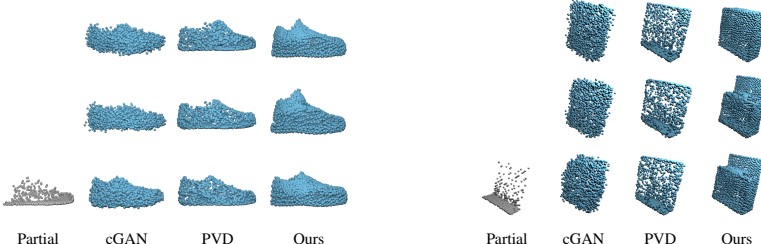

Partial    cGAN    PVD    Ours        Partial    cGAN    PVD    Ours

Figure 6: Qualitative comparison of diverse completions on the Google Scanned Objects dataset.

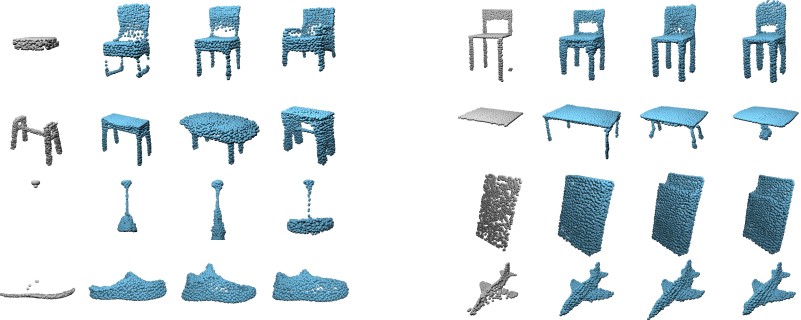

Figure 7: Multi-modal completions (blue) of partial point clouds (gray) produced by our method.

Additionally, we compare our method on real data from the Google Scanned Objects dataset. In Table 3, we show that our method obtains better performance across all the metrics for all three categories. We present a qualitative comparison of completions on objects from the Google Scanned Objects dataset in Figure 6. Completions by cGAN [8] are noisy and lack diversity, while completions from PVD [10] have little diversity and suffer from non-uniform density. Alternatively, our method produces cleaner and more diverse completions with more uniform density.

We further demonstrate the ability of our method to produce diverse high-quality shape completions in Figure 7. Even with varying levels of ambiguity in the partial scans, our method can produce plausible multi-modal completions of objects. In particular, we find that under high levels of ambiguity, such as in the lamp or shoe examples, our method produces more diverse completions. On the other hand, when the object is mostly observed, such as in the plane example, our completions exhibit less variation among them. For more qualitative results, we refer readers to our supplemental.

Finally, we find that our method is capable of inference in near real-time speeds. To produce $K = 10$ completions of a partial input on a NVIDIA V100, our method takes an average of $85$ ms while cGAN and KNN-latent require $5$ ms. PVD requires $45$ seconds which is $500$ times slower than us.

Table 4: Ablation on style code dim.

| Style Code | MMD ↓ | TMD ↑ | UHD ↓ |
|---|---|---|---|
| 512-dim | 1.51 | 3.41 | 3.98 |
| 128-dim | 1.54 | 3.30 | 4.17 |
| 32-dim | 1.59 | 3.89 | 4.04 |
| 16-dim | 1.55 | 3.96 | 3.97 |
| 8-dim | 1.51 | 3.94 | 3.97 |
| 4-dim | 1.51 | 4.00 | 3.93 |
| 8-dim + noise (Ours) | 1.50 | 4.36 | 3.79 |

Table 5: Ablation on style code generation.

| Method | MMD ↓ | TMD ↑ | UHD ↓ |
|---|---|---|---|
| Gaussian Noise | 1.57 | 3.71 | 4.59 |
| Mapping Network | 1.45 | 4.03 | 3.72 |
| Style Encoder (Ours) | 1.50 | 4.36 | 3.79 |

Table 6: Ablation on completion network design.

| Method | MMD ↓ | TMD ↑ | UHD ↓ |
|---|---|---|---|
| SF | 1.45 | 3.21 | 3.37 |
| SF + PE | 1.56 | 3.74 | 3.83 |
| SF + PE + PCI (Ours) | 1.50 | 4.36 | 3.79 |

Table 7: Ablation on discriminator architecture.

| Method | MMD ↓ | TMD ↑ | UHD ↓ |
|---|---|---|---|
| Single-scale | 1.58 | 2.05 | 4.27 |
| Multi-scale (Ours) | 1.50 | 4.36 | 3.79 |

Table 8: Ablation on loss functions.

| | MMD ↓ | TMD ↑ | UHD ↓ |
|---|---|---|---|
| w/o $\mathcal{L}_{comp}$ | 1.62 | 4.61 | 4.58 |
| w/o $\mathcal{L}_{part}$ | 1.81 | 5.97 | 13.03 |
| w/o $\mathcal{L}_{div}$ | 1.70 | 0.41 | 3.57 |
| Ours | 1.50 | 4.36 | 3.79 |

## 4.2 Ablation studies

In Table 4 we examine the dimensionality of our style codes. Our method obtains higher TMD when using smaller style code dimension size. We additionally find that adding a small amount of noise to sampled style codes during training further helps boost TMD while improving UHD. We believe that reducing the style code dimension and adding a small amount of noise helps prevent our style codes from encoding too much information about the ground truth shape, which could lead to overfitting to the ground truth completion. Furthermore, in Table 5, we present results with different choices for style code generation. Our proposed style encoder improves diversity over sampling style codes from a normal distribution or by using the mapping network from StyleGAN [46]. Despite having slightly worse MMD and UHD than StyleGAN's mapping network, we find the quality of completions at test time to be better when training with style codes sampled from our style encoder (see supplementary).

In our method, we made several changes to the completion network in SeedFormer [7]. We replaced the encoder from SeedFormer, which consisted of point transformer [61] and PointNet++ [62] set abstraction layers, with our proposed partial encoder as well as replaced the inverse distance weighted interpolation with PointConv interpolation in the SeedFormer decoder. To justify these changes, we compare the different architectures in our GAN framework in Table 6. We compare the performance of the original SeedFormer encoder and decoder (SF), our proposed partial encoder and SeedFormer decoder (PE + SF), and our full architecture where we replace inverse distance weighted interpolation with PointConv interpolation in the decoder (PE + SF + PCI). Our proposed partial encoder produces an improvement in TMD for slightly worse completion fidelity. Further, we find using PointConv interpolation provides an additional boost in diversity while improving completion fidelity.

The importance of our multi-scale discriminator is shown in Table 7. Using a single discriminator/diversity penalty only at the final output resolution results in a drop in completion quality and diversity when compared with our multi-scale design.

Finally, we demonstrate the necessity of our loss functions in Table 8. Without $\mathcal{L}_{comp}$, our method has to rely on the discriminator alone for encouraging sharp completions in the missing regions. This leads to a drop in completion quality (MMD). Without $\mathcal{L}_{part}$, completions fail to respect the partial input, leading to poor UHD. With the removal of either of these losses, we do observe an increase in TMD; however, this is most likely due to the noise introduced by the worse completion quality. Without $\mathcal{L}_{div}$, we observe TMD drastically decreases towards zero, suggesting no diversity in the completions. This difference suggests how crucial our diversity penalty is for preventing conditional mode collapse. Moreover, we observe that when using all three losses, our method is able to obtain good completion quality, faithfully reconstruct the partial input, and produce diverse completions.

## 5 Conclusion

In this paper, we present a novel conditional GAN framework that learns a one-to-many mapping between partial point clouds and complete point clouds. To account for the inherent uncertainty present in the task of shape completion, our proposed style encoder and style-based seed generator enable diverse shape completion, with our multi-scale discriminator and diversity regularization preventing mode collapse in the completions. Through extensive experiments on both synthetic and real datasets, we demonstrate that our multi-modal completion algorithm obtains superior performance over current state-of-the-art approaches in both fidelity to the input partial point clouds and completion diversity. Additionally, our method runs in near real-time speed, making it suitable for applications in robotics such as planning or active perception.

While our method is capable of producing diverse completions, it considers only a segmented point cloud on the object. A promising future research direction is to explore how to incorporate additional scene constraints (e.g. ground plane and other obstacles) into our multi-modal completion framework.

## Acknowledgments and Disclosure of Funding

This work was supported in part by ONR award N0014-21-1-2052, ONR/NAVSEA contract N00024-10-D-6318/DO#N0002420F8705 (Task 2: Fundamental Research in Autonomous Subsea Robotic Manipulation), NSF grant #1751412, and DARPA contract N66001-19-2-4035.

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
