# OpenReview forum: "Diverse Shape Completion via Style Modulated Generative Adversarial Networks"
_NeurIPS.cc/2023/Conference — NeurIPS 2023 poster_

### Official Review · Reviewer_Ahkf · 2023-06-28

**Soundness:** 3 good
**Presentation:** 2 fair
**Contribution:** 2 fair
**Rating:** 5
**Confidence:** 4

**Summary:**

This paper proposes a new conditional generative network that can produce diverse completions of a partially observed point cloud. The stochasticity is introduced via style modulation. A style code is learned to explicitly carry shape category information leading to better completions. Moreover, diversity penalties and discriminators at multiple scales are also set to prevent conditional mode collapse. Experiments show that the proposed framework can achieve significant improvements in respecting the partial observations while obtaining greater diversity in completions.

**Strengths:**

The results on different datasets show the effectiveness of the proposed method.

**Weaknesses:**

1.	The main target of this paper is to achieve diverse synthesis. However, no such visual samples are provided. The authors should provide the diverse synthesis for different categories in the main paper.

2.	How to decide the optimal values for loss weights in Eq. 7? Although the authors claim that the same loss weight setting can lead to good results on all experimental datasets, the experimental datasets only contain several categories. More categories should be considered into analysis.

3.	The idea of this paper is derived from the 2D StyleGAN. The difference about the idea should be clearly indicated.

4.	The method has not compared with SOTA methods that use the diffusion model, like [30, 31]. And this method should also support unconditional point cloud synthesis?


**Questions:**

Please answer the questions in the weakness section.
Moreover, can this method be utilized into the point cloud completion of scenes? Like ScanNet dataset.

**Limitations:**

Further considering point cloud completion with text conditions can help to achieve open-world and diverse synthesis effects, which can improve the contribution of this paper.

---

> ### Author Rebuttal · Authors · 2023-08-09
>
> We thank reviewer Ahkf for their valuable and constructive comments.
>
> **No visual samples of diverse synthesis are shown**
> Qualitative comparisons to other methods were included in Figures 1, 4 and 5 in our main paper. In each of these figures we show 3 completions produced for the same partial input. We additionally shared more visual examples in our supplemental material. We will add a sentence to our main paper mentioning that more visual examples can be found in our supplementary material.
>
> **How did you decide optimal values for loss weights? More categories should be considered for analysis.**
> The loss weights were set by initially setting them all to a value of 1. We found this led to overfitting to the single ground truth completion and less diverse completions, thus we decreased the weight of our completion loss and increased the weight of our diversity penalty, which led to more satisfactory results.
>
> In terms of the categories chosen, we follow previous works (cGAN [8], IMLE [9], PVD [10], ShapeFormer [13]) and evaluate on the chair, table, and airplane categories of the 3DEPN dataset and the chair, table, and lamp categories of the PartNet dataset. To further validate our method we have performed additional experiments on the shoe, toys, and consumer goods categories of the Google Scanned Objects dataset.
>
> **Idea is derived from 2D StyleGAN [54], the difference should be clearly indicated.**
> Note that the only idea we use from StyleGAN is weight modulation as a way to inject style codes into our features. 2D StyleGAN is focused on unconditional 2D image generation while our work is on diverse 3D shape completion. StyleGAN starts from a learned constant tensor while our work starts from a set of features extracted from a partial shape. Additionally, in StyleGAN, style codes start from random noise and are mapped to style space through a mapping network while in our work we extract style codes during training from complete shapes. Please refer to lines 151-158 in our paper for a discussion on StyleGAN's style code versus ours as well as lines 180-185 where we mention partial input features and learned style codes as input to our style modulation.
>
> **Method does not compare against SOTA diffusion models [30, 31]. Does the method support unconditional point cloud synthesis?**
> The works by Luo et al. [30] and Zeng et al. [31] are SOTA methods for the task of unconditional 3D shape generation while our work is focused on the task of diverse point cloud completion. Their methods do not accept a partial point cloud as input and hence cannot be compared on the task we're trying to solve.
>
> We have compared against Point-Voxel Diffusion (PVD) by Zhou et al. [10] which is a diffusion based method that can perform diverse shape completion. We show our method outperforms PVD quantitatively and qualitatively in Table 2, Table 3, and Figure 5 of our main paper as well as Figure 3 of our supplemental. We also find our method runs 500 times faster than PVD.
>
> Our method does not directly support unconditional shape generation as our seed generator expects partial shape features as one of its inputs. The seed generator could be modified to be conditioned on random noise instead of partial shape features; however, that is outside the scope of this paper which is point cloud completion.
>
> **Can the method be used on datasets like ScanNet?**
> Our method works on segmented point clouds and thus it is possible to work on ScanNet if segmented properly. However, our method would need to be modified to avoid producing completions that intersect with other occupied points in the scene, which we mentioned as future work in lines 328-329 of our main paper.

---

> > ### Comment · Reviewer_Ahkf · 2023-08-19
> >
> > Thank you for the rebuttal from the authors. The answers from the authors are reasonable, and I decided to keep the positive score.

---

### Official Review · Reviewer_R39m · 2023-07-03

**Soundness:** 3 good
**Presentation:** 3 good
**Contribution:** 3 good
**Rating:** 6
**Confidence:** 4

**Summary:**

The paper proposes to reconstruct partial point cloud inputs using a multi-modal process, where the generator can output multiple plausible shapes. The key idea is to have a separate network (StyleEncoder) that extracts style from an input, in addition to a separate network (PartialEncoder) that extracts structural information, that are then combined to generate the final reconstruction. Another contribution is a multi-scale discriminator that checks real and fake pairs at multiple generator output stages. The entire pipeline is essentially a conditional GAN network for generating 3D shapes.

**Strengths:**

The paper proposes to address an important problem where there could be multiple potential reconstructions given a partial input. Limitations of recent work (SeedFormer [7]) are addressed with a clever network design strategy. The method is evaluated on three public datasets (3D-EPN, GSO, and Part-Net) with promising quantitative results.

**Weaknesses:**

The methodology section is difficult to follow, and I found it hard to understand the training process. In addition, it seems that the proposed technique is heavily based on previous work (e.g., SegFormer). It may help to explain the key technical contributions with respect to these previous works and defer the reader to the other papers for understanding technical details.

In addition, the validation of the proposed technique is rather limited. The training procedure, and validation steps are not clearly described. Is the method sensitive to shape orientation? What type of data augmentation is used during training, and is it trained on a per-category level (thus learning certain shape priors) or category-agonostic level?

I would have expected to see ablation experiments with same methodology but different network architecture. Finally, the authors evaluate on PartNet, yet, never quantify part level accuracy.

In Tables 5 and 6, ablation studies of the style code and discriminator have very similar MMD metric but large UHD metric. I did not understand the choice and motivation behind these metrics. Why not use standard Hausdorff distance instead?

Finally, the method is geared toward generative multiple outputs given a single input, yet this capability is never evaluated. I would have expected to see additional examples where multiple plausible reconstructions an input can have, followed by a user study or a qualitative evaluation of the results.

**Questions:**

1.	On line 256, it is stated the input contains 1024 points and the output contains 2048 points. Why is there a mismatch in these numbers?
2.	If the input is meant to be a partial point cloud, have the authors considered the effect of point density?
3.	Are selected datasets benchmark datasets for partial shape reconstruction?
4.	Is the goal of the multi-scale discriminator to address partial input (lack of ground truth for each structure/style pair, or is it to address structure at different scales)? Given the name and design, the naming and purpose is confusing.


**Limitations:**

The paper proposed clever modifications on an existing network (SegFormer) to allow it to generate multiple completions given a single partial input. The idea is general, but it does not appear that the method is well-validated outside of the specific architecture considered, and thus, its impact is not clear.

---

> ### Author Rebuttal · Authors · 2023-08-10
>
> We thank reviewer R39m for their valuable and constructive comments.
>
> **Paper is difficult to follow. Helps to state key contributions.**
> We'd like to clarify any confusions you've had. Meanwhile, note that 2 other reviewers found the paper well-written. First, we want to note that our work is not related to SegFormer, but is related to the deterministic point cloud completion method SeedFormer [7].
>
> Our main contributions are outlined in lines 63-69 of our paper. In particular, we propose:
> - A diverse seed generator which produces diverse completions via style modulation of partial shape features (see lines 179-182)
> - To learn the style codes used in style modulation from the distribution of complete shapes via a style encoder (see lines 151-158)
> - Discriminators at multiple scales to enable shape completion without access to multiple ground truth completions per partial input (see lines 205-207 and 217-218)
> - A diversity penalty at multiple scales which prevents conditional mode collapse (see lines 226-232)
>
> None of these exist in SeedFormer, which is completely deterministic in the completion it generates.
>
> **Validation is limited. Training and validation is not clearly described. Data augmentation performed?**
> We beg to differ about the claim that the validation is limited. We have quantitatively evaluated on 3 different datasets, compared against major baselines including a diffusion model, and performed multiple ablation studies. Training procedure, datasets, and metrics used can be found in lines 256-273 of our paper. We provided full dataset information and mathematical definitions of metrics in Sections 3 and 4 of our supplemental.
>
> We did not perform any data augmentation to fairly compare against prior methods which also did not perform any augmentation. Our sensitivity to shape orientation is hence similar to prior work.
>
> Our method is trained per category similar to other diverse shape completion works we compare against [8, 9, 10].
>
> **Expected to see ablation experiments on architecture. Part level accuracy was not quantified for PartNet**
> We provide additional ablations on network design in our supplemental, where we compare our PointConv-based partial encoder with SeedFormer's partial encoder in Table 1 and compare our proposed diversity penalty with an alternative baseline diversity penalty in Table 2.
>
> We have included a new ablation on our losses in the table presented in our global response.
>
> We do not evaluate part level accuracy in PartNet because for the task of multimodal shape completion, it is not required to completely reconstruct all the ground truth shapes. Also, it is not possible to evaluate bidirectional metrics such as Chamfer distance or MMD as completions are not predicted at the part level; thus there is no notion of what part a point belongs to. Additionally, one-side metrics such as UHD would make incorrect penalizations as not all parts need to exist in the completions to be considered valid. For example, a ground truth chair may have arm rests; however, if a partial point cloud of the chair only contains its legs, completions of the chair without arm rests may still be valid completions. Also, note such a metric is not computed in any of the works we compare against [8, 9, 10, 13].
>
> **Why do ablation results have similar MMD but large UHD? What is the motivation of these metrics? Why not standard Hausdorff distance?**
> The UHD we used is the standard one-sided Hausdorff distance. MMD performs an average over distance to nearest neighbors while UHD perform a max over them and thus is more susceptible to large variations in the presence of noisy completions (see Section 4 of supplemental for mathematical definitions of metrics).
>
> MMD measures completion quality and coverage of the test set and UHD measures fidelity to the partial input. These are standard evaluation metrics for the task of diverse shape completion [8, 9, 10, 13].
>
> **Method is geared toward generating multiple outputs but capability is never evaluated.**
> Note that the TMD metric quantitatively evaluates the diversity of generated outputs. Besides, qualitative examples of diverse completions are shown in Figures 1, 4 and 5 of our main paper. In the supplemental, we shared more visual examples of our method producing diverse completions per partial input. We will add a sentence to our main paper mentioning that more visual examples can be found in our supplemental.
>
> **Why does input have 1024 points and output have 2048 points?**
> We follow the standard convention for multimodal shape completion which is to use a 1024 point partial input and produce a 2048 point completion as output [8, 9, 10, 13].
>
> **Is input point density considered?**
> We make no assumption on point density as it can vary heavily across shapes and viewpoints. 3DEPN and GSO inputs are generated by lifting depth maps into 3D and thus exhibit non-uniform density. For PartNet, we uniformly sample points in the parts kept; however, the number of parts and size of each part can still vary.
>
> **Benchmark datasets for diverse completion task?**
> 3DEPN and PartNet datasets are benchmark datasets for diverse shape completion that previous works have evaluated on [8, 9, 10, 13]. We also evaluated on Google Scanned Objects to see how our method performs when trained with real data.
>
> **Goal of multi-scale discriminator?**
> We introduce adversarial learning to address lack of ground truth for each structure/style pair (lines 205-207 of our paper).
>
> The goal of our multi-scale discriminator is to discriminate completions at different output resolutions. At coarser levels, points represent the skeletal structure of the shape, with each point being more important to whether the shape looks realistic. We find that discriminating at these coarse levels along with the finer levels helps with shape completion. We show a completion result using single vs. multiple discriminators in Figure 5 of our supplemental.

---

> > ### Comment · Reviewer_R39m · 2023-08-15
> > **Thank you**
> >
> > Thank you for providing a detailed rebuttal. I have misunderstood parts of the paper and the rebuttal helped clarify, therefore, I increased my score. In the method section, please focus on the contributions of the approach, just like you did in the rebuttal , so it is easy to decipher which part is novel and what is borrowed from prior work. A potential and partial solution is to rename subsections to include key ideas (e.g., style encoder via style modulation).
> >
> > The sentence on line 157-157 that begins with "our style codes explicitly carry style information about a shape category.." is confusing since you only train per category, not across categories.
> >
> > In figures 1,4,5 where you include multiple completions, please add labels to explain that each (blue) shape is a possible completion.
> >
> > Information about train/val/test splits should be included in the main manuscript, not in the supplementary material.

---

> > > ### Author Response · Authors · 2023-08-20
> > >
> > > Thank you for the feedback and suggested changes. We will do our best to incorporate these changes in the final version of our paper.

---

### Official Review · Reviewer_ohCS · 2023-07-04

**Soundness:** 3 good
**Presentation:** 3 good
**Contribution:** 2 fair
**Rating:** 5
**Confidence:** 5

**Summary:**

The paper proposes a diverse shape completion method by extracting style codes from complete shapes and learning a distribution over them. Moreover, diversity penalties and discriminators at multiple scales are introduced as well to prevent conditional modal collapse to generate various object shapes. To verify the effectiveness of the method, various experiments are conducted, and promising results are observed.

**Strengths:**

1. The paper is well written and organized.
2. The methods are interesting and promising results are obtained.

**Weaknesses:**

1. According to line 156, “style codes” carrying category information are learnt from the distribution of complete shapes, however, the goal is to generate diverse shapes not only among different classes but also within the same class. Hence, how to get the diverse information for a single class seems missing in the paper.
2. What is the difference of style encoder technique compared to VAE? According to the illustration in Sec. 3.2, it looks the same to VAE, which potentially results in limited novelty of the proposed style encoder.
3. It would be better to also show some ablation studies on the network architectures by removing specific components to see how different parts contribute to the overall performances, which could be done by settings specific loss's weight to 0.

**Questions:**

None.

**Limitations:**

Some limitations are discussed in detail. To make a more comprehensive comparison with other SOTA works, possible memory consumption or time efficiency could be illustrated.

---

> ### Author Rebuttal · Authors · 2023-08-10
>
> We thank reviewer ohCS for their valuable and constructive comments.
>
> **Diverse information for a single class seems to be missing.**
> Our method is trained per category similar to other multimodal shape completion works such as cGAN [8], IMLE [9], and PVD [10]. Thus, the "style codes" we learn carry different style information from complete shapes within a category rather than across categories. In Figure 2 of our attached rebuttal PDF we project style codes to 2D space using PCA and show that neighborhoods in 2D space contain shapes with similar characteristics/styles.
>
> **What is the difference of style encoder technique from VAE? It looks similar to VAE and thus there is potentially limited novelty of the style encoder.**
> In principle, the style encoder uses similar loss functions and sampling tricks as VAE. However, there is a significant difference as we seek to make the reconstruction capability from the style encoder **worse** by using less dimensions and an inefficient PointNet encoder. The reason is that the input of the style encoder is the ground truth shape and if one reconstructs it directly from the style encoder, then the encoding from the partial input would have little purpose, which hurts generalization and diversity. Table 4 shows our finding that by making the style encoder **worse** by using less dimensions and adding noise, we were able to take more information from the other partial input encoder and achieve better diversity and reconstruction error during test time. We believe this is a novel finding.
>
> **Would be better to show some ablations on network architecture by removing components/losses**
> In our global response we have provided a new table where we train our method with each of $L_{comp}$, $L_{div}$, and $L_{part}$ set to 0. Please see our global response for further comments on this ablation.
>
> **Would be nice to show memory consumption or time efficiency**
> In the last paragraph of Section 4.1 (lines 297-299) of our main paper we provide a comparison to other methods of average inference speed for producing 10 completions of a single partial input. We find that our method is capable of near real-time speeds and is 500 times faster than diffusion-based method PVD [10].

---

> > ### Comment · Reviewer_ohCS · 2023-08-15
> >
> > Thanks for the rebuttal, and most of my concerns are addressed. Hence, I would like to keep my positive rating. One minor concern is that TMD and UHD in ablation studies actually show a bit worse performances when overall losses are used.

---

> > > ### Author Response · Authors · 2023-08-21
> > >
> > > Thank you for the response. We hope we can address the minor concern you have.
> > >
> > > While the TMD is slightly worse when using all our losses, this does not necessarily correspond to worse diversity. As we mention in our global rebuttal, TMD is easy to increase when completion quality (MMD) and fidelity to the partial input (UHD) are poor, as noisy completions are one way to produce high TMD. Note that the two cases where TMD is higher (w/o $L_{comp}$ or w/o $L_{part}$), the MMD and UHD are worse suggesting that the TMD may be driven up through noise rather than more diverse completions.
> > >
> > > In regards to UHD, we make a trade off that we believe is worthwhile. When using all our losses compared to w/o $L_{div}$, we sacrifice performance on UHD slightly for a large increase in TMD. We would also like to point out that the UHD we obtain with all our losses still significantly outperforms the UHD obtained by other multimodal shape completion methods we compare against.

---

### Official Review · Reviewer_iNun · 2023-07-05

**Soundness:** 3 good
**Presentation:** 3 good
**Contribution:** 3 good
**Rating:** 6
**Confidence:** 4

**Summary:**

The goal of multimodal shape completion is to generate many different plausible completions of an incomplete shape. Based on the conditional GAN, this paper introduces two key concepts to improve the diversity and accuracy of multimodal completion. One is to use style codes instead of random noise, which means better distribution of complete shapes. The other is the use of multiscale discriminators to refine the predicted shapes from coarse to fine. The method is evaluated on both synthetic and real data sets and shows significant improvements over comparable methods based on standard metrics.

**Strengths:**

1. The results of multimodal completion look very clean and diverse. The numerical results show a new SOTA.

2. The use of the learned style codes instead of Gaussian noise is reasonable since the former is more similar to the distribution of complete shapes.

**Weaknesses:**

1. Ablation study with and without $\mathcal L_{comp}$ and $\mathcal L_{div}$ is not provided.

2. The flat and thin structures of the shapes generated by the proposed method look good, while other methods are more noisy. For the task of multimodal completion, this is quite impressive.
According to Figure 5 in the supplementary file, the main reason is probably the use of multiscale discriminators, since the results of a single scale are noisy. Is this true? And why are 4 scales used? If more than 4 scales are used, is the quality continuously improved?

3. The distribution of style codes should better reflect the actual shape distribution. To better verify this, a visual analysis comparing the distribution of the different conditional codes is helpful.

**Questions:**

1. Is this method applicable to unseen categories?

**Limitations:**

The limitation is well discussed.

---

> ### Author Rebuttal · Authors · 2023-08-09
>
> We thank reviewer iNun for their valuable and constructive comments.
>
> **Ablation study on $L_{comp}$ and $L_{div}$**
> In our global response we have provided a new table where we train our method with each of $L_{comp}$, $L_{div}$, and $L_{part}$ set to 0. Please see our global response for further comments on this ablation.
>
> **Is the multi-scale discriminator the reason for sharp/clean completions? Why are 4 scales used?**
> Yes, we find that when using a single global discriminator it is not able to effectively discriminate against real and fake point clouds and hence we use several of them at different point cloud resolutions/scales. We tried out more complicated discriminators involving PointConv and attention mechanisms but were not able to successfully train with such methods, which is in line with the findings of Wang et al. [55] (see also lines 214-218 of our paper).
>
> We also find our loss functions $L_{comp}$ and $L_{part}$ to be important for producing sharper completions (see our new table and comments on this presented in our global response).
>
> We use 4 scales as our method makes use of the upsampling procedure from the SOTA deterministic point cloud completion method SeedFormer [7], which also uses 4 layers (an initial coarse completion + 3 upsampled completions). Note we start with a coarse completion of 256 points and upsample by a factor of 2 at each upsampling layer to produce a final completion of 2048 points. We did not explore using more than 4 scales since if we introduce more scales past 2048 points we would not be able to fairly compare against all the other multimodal shape completion methods which produce 2048 points as output.
>
> **Visualization of distribution of style codes**
> In Figure 2 of our attached rebuttal PDF we share a plot of our learned style codes projected into 2D space using PCA. By querying random 2D points and visualizing the corresponding ground truth shapes in the neighborhood we find that the shapes in each cluster/neighborhood tend to share some characteristic/style. Such a result suggests that our style encoder is effectively learning to extract styles from the distribution of complete shapes.
>
> **Is method applicable to unseen categories?**
> No, our method is trained per-category similar to cGAN [8], IMLE [9], and PVD [10].

---

### Official Review · Reviewer_A3pQ · 2023-07-07

**Soundness:** 3 good
**Presentation:** 3 good
**Contribution:** 3 good
**Rating:** 7
**Confidence:** 3

**Summary:**

This work proposes a novel GAN for diverse shape completion from partial point clouds. To enable diverse completions, a style-based generator is introduced that leverages style codes from a learned distribution of complete shapes for style modulation. Further, a multi-scale discriminator and a diversity penalty are proposed for better diversity.

**Strengths:**

The paper is well-written and easy to follow. The experimental evaluation is thorough and supports the main claims in the paper. Both quantitative and qualitative results demonstrate the efficiency of the proposed approach over existing works.


**Weaknesses:**

While the supplementary includes an ablation with different diversity penalties, it should also contain an experiment without the diversity penalty for completeness. As the diversity penalty is one of the contributions of this work, I would consider moving the study to the main paper.

It would further be interesting to report the nearest neighbors in the training set for the completed regions in the results because the style encoder learns a distribution over the completed shapes.

A discussion of failure cases should be added to the supplementary.

For completeness, the missing values in Table 3 for “Ours \dagger” should be added.


**Questions:**

Please also see Weaknesses.

How can diffusion-based methods, e.g. LION, be extended for point cloud completion? Would it be possible to train a variant that was conditioned on the partial point cloud (i.e. similar to 2D image inpainting with diffusion models) and report this as a further baseline?
While diffusion models are limited wrt to inference speed, it would be interesting to include them as a baseline due to their strong generative performance.

**Limitations:**

The limitations were discussed but a broader impact session should be added.

---

> ### Author Rebuttal · Authors · 2023-08-09
>
> We thank reviewer A3pQ for their valuable and constructive comments.
>
> **Diversity penalty ablation**
> In our global response we have provided a new table where we train our method without diversity penalty $L_{div}$. Please see our global response for further comments on this ablation. We'll adopt our diversity penalty ablations to the main paper in the final version.
>
> **Report nearest neighbors in the training set for completions**
> In Figure 1 of our attached rebuttal PDF we have included some examples of the nearest neighbor in the training set to a completion. Please see our global response for further comments on this. We will add this to our supplementary material in our final version.
>
> **Discussion of failure cases**
> We have included a few examples of failure cases in Figure 3 of our attached PDF for our rebuttal. Similar to any other generative model, our method occasionally fails to produce plausible completions. We observe that the occasional failures that happen are usually either due to missing thin structures or some noisy artifacts. We will add this to our supplemental in our final version.
>
> **Missing values in Table 3**
> Note that for the methods denoted by $\dagger$ in Table 3, we evaluate using the protocol of PVD [10]. In particular, we produce a completion with 2048 points, subsample 1024 of them, and then directly concatenate the subsampled 1024 points to the partial input. In this case, it is not useful to evaluate UHD as it will always be 0 and thus we simply mark it with an asteriks.
>
> **Extending LION [31] to point cloud completion as a baseline to compare against diffusion models**
> While modifying unconditional shape generation methods like LION to handle point cloud completion could be done, such a modification would require design/architectural choices on how to incorporate the partial point cloud input. This would most likely take a significant amount of effort to produce a reasonable baseline. Instead, we compare our method to Point-Voxel Diffusion (PVD) [10] which is a diffusion-based method that can perform diverse shape completion. We show our method outperforms PVD quantitatively and qualitatively, meanwhile being 500 times faster than it, in Table 2, Table 3, and Figure 5 of our main paper as well as Figure 3 of our supplementary material.
>
> **Broader impacts**
> As a point cloud completion method, it has the potential to be used in robotics applications such as planning. As of right now, we do not foresee immediate negative social impacts until our method is integrated into real robots. However, incorporating such a completion method into robotics applications is a scope for future work.

---

> > ### Comment · Reviewer_A3pQ · 2023-08-15
> >
> > Thank you for the rebuttal and the additional ablation study. This answers my questions.

---

### Author Rebuttal · Authors · 2023-08-09

We thank the reviewers for taking the time to read and review our work. We have tried our best to answer and address any questions and clarifications in each individual reviewer response. In the rest of our global response, we discuss a common ablation that was requested across several reviewers and share some comments on some additional figures in the PDF we have attached for our rebuttal.

--------------------------
### Ablation of loss functions:

|     | MMD ($\downarrow$) | TMD ($\uparrow$)     | UHD ($\downarrow$)    |
| :---        |    :----:   |    :----:   |         ---: |
| w/o $L_{comp}$      | 1.62         | 4.61      |4.58    |
| w/o $L_{part}$    | 1.81      | 5.97       | 13.03       |
| w/o $L_{div}$    | 1.70         | 0.41        | 3.57      |
| Ours   | 1.50       | 4.36        | 3.79       |


As requested by several reviewers, we have performed an ablation on our loss functions in the table above. Without $L_{comp}$ our method has to rely on the discriminator alone for encouraging sharp completions in the missing regions. This leads to a drop in completion quality (MMD). It also hurts partial reconstruction quality (UHD) as our completion loss $L_{comp}$ provides further gradient signal to the partially observed regions as well. Without $L_{part}$, completions fail to respect the partial input, leading to poor UHD. We observe that failing to respect the partial input also leads to a general degradation in completion quality (worse MMD). With the removal of either of these losses, we do observe an increase in TMD; however, we note that TMD is trivial to increase with worse completion quality as noise can be simply introduced. Without $L_{div}$, we observe TMD drastically decreases towards 0, suggesting no diversity in the completions. This difference suggests how crucial our diversity penalty is for preventing conditional mode collapse. We observe that when using all three losses, our method is able to obtain good completion quality, faithfully reconstruct the partial input, and produce diverse completions. We'd include this study in the final version.

--------------------------
### Visualizations in attached PDF:
In Figure 1 of our attached PDF, we share several completions (in blue) of a partial input and each completions nearest neighbor (in yellow) to a ground truth complete shape in the training set. Note that our method produces a different nearest neighbor for each completion of a partial input, showing that our method can overcome conditional mode collapse. Additionally, each nearest neighbor is similar to the partially observed region and varies more in the missing regions, suggesting that our method is capturing plausible diversity in our completions that matches with variance in the ground truth shape distribution.

In Figure 2 of our attached PDF, we plot our learned style codes from shapes in the training set by projecting them into 2D using principal component analysis (PCA). To better understand whether our style encoder is learning to extract style from the shapes, we visualize the corresponding shapes in random neighborhoods/clusters of our projected data. We find that the shapes contained in a neighborhood have a shared style or characteristic. For example, the chairs in the brown cluster all have backs whose top is curved while the black cluster has chairs that all have thin slanted legs.

---

### Decision · Program_Chairs · 2023-09-21

**Decision:**

Accept (poster)

**Comment:**

The reviews of the paper is overall positive. This paper proposes a novel GAN-based method to generate diverse completion from partial point clouds. The diversity is demonstrated to outperform prior methods on multiple datasets. The rebuttal has addressed most of the comments. Thus I recommend acceptance. Also, I encourage the authors to revise the paper based on reviewers' suggestions and as promised in the rebuttal.